## Comment

health and disease and epidemiology

**Author for correspondence:**
M. J. Plank
e-mail: michael.plank@canterbury.ac.nz

# Comment: weekly COVID-19 testing with household quarantine and contact tracing is feasible and would probably end the epidemic

## M. J. Plank[1,2], A. James[1,2] and N. Steyn[2,3]

[1]School of Mathematics and Statistics, University of Canterbury, Christchurch, New Zealand
[2]Te Pūnaha Matatini, Centre of Research Excellence in Complex Systems, New Zealand
[3]Department of Physics, University of Auckland, New Zealand

MJP, 0000-0002-7539-3465; AJ, 0000-0002-1543-7139;
NS, 0000-0001-8904-2941

## 1. Introduction

Peto *et al.* [1] suggest that weekly testing of the entire UK population for SARS-CoV-2 combined with household quarantine could rapidly end the spread of COVID-19. The logistical challenges in conducting and processing 10 million tests per day over an extended period are formidable [2]. Even if these challenges could be overcome, it is not obvious that this strategy would reduce the effective reproduction number $R_{eff}$ sufficiently to control the epidemic. Peto *et al.* [1] support their claim with a statistical footnote. However, this does not actually calculate the proportion of transmissions that would be prevented, nor allow for high false-negative rates before symptom onset [3]. They also assume that one-third of transmissions are within households, which is likely to be an overestimate (see below).

We use a simple household model of COVID-19 transmission (figure 1a), combined with data on reverse transcriptase polymerase chain reaction (RT-PCR) test sensitivity [3], to estimate the reduction in $R_{eff}$ from mass weekly testing and household quarantine. Even with optimistic assumptions about the test processing time (1 day) and the effectiveness of home isolation and quarantine (100% effective), this strategy would only reduce $R_{eff}$ by around 21%. If $R_0 = 2.5$ in the absence of interventions, mass testing reduces $R_{eff}$ to 2.1 and the addition of household quarantine further reduces $R_{eff}$ to 2.0. Under more realistic parameter settings, such as partially effective quarantine [5] or incomplete testing coverage, $R_{eff}$ would be larger than this. Rapid antigen tests can provide faster turnaround of test results. While this may appear to be a solution to some of the problems outlined here, removing the assumed 1 day

**Figure 1.** (*a*) Household branching process model for COVID-19 transmission. Each household (blue ellipses) is randomly assigned a number of members between 1 and 8 according to the distribution of household sizes for England and Wales [4] (2011 census, all usual residents aged 18 and over in households). Households are numbered in order of the time of first infection; only infected individuals and their transmission routes are shown in the diagram. Once a household is placed in quarantine, all further onward transmission from all individuals in that household is prevented. (*b*) Effective reproduction number $R_{eff}$ with no control (blue), under mass testing (red) and mass testing plus household quarantine (yellow), and mass testing, household quarantine and instant return of test results ($t_{test} = 0$, purple) for varying proportions of household transmission. This was done by varying the household secondary attack rate $SAR_{home}$ between 0% (no household transmission) and 70% (33% of all transmission is within households) while fixing $R_0 = 2.5$. Vertical dotted line shows the results when household secondary attack rate is 20%. Results are averaged over 200 independent realizations of the model, each initialized with one infected seed case and run for 30 days.

**Table 1.** Effective reproduction number $R_{eff}$ under different control measures and with differing levels of individual heterogeneity in transmission rates (Poisson distribution has no heterogeneity, negative binomial distributions with smaller values of *k* have more heterogeneity). Results are for $SAR_{home} = 0.2$ and are averaged over 500 independent realizations of the model, each initialized with one infected seed case and run for 30 days.

| | no control | mass testing | mass testing + HHQ | mess testing + HHQ + instant |
|---|---|---|---|---|
| *Poisson ($R_{out}$)* | 2.49 | 2.12 | 2.01 | 1.78 |
| *NegBin ($R_{out}$, k = 1)* | 2.50 | 2.11 | 1.99 | 1.77 |
| *NegBin ($R_{out}$, k = 0.5)* | 2.51 | 2.13 | 2.01 | 1.78 |
| *NegBin ($R_{out}$, k = 0.1)* | 2.49 | 2.09 | 2.05 | 1.82 |

delay from test to result only reduces $R_{eff}$ to 1.8. In addition, these tests typically have lower sensitivity than RT-PCR tests [6], and relying solely on self-testing may compromise adherence to isolation and quarantine.

The results for $R_{eff}$ above assume that the household secondary attack rate ($SAR_{home}$) is 20%, which means that around 12% of transmission is within households. This is at the upper end of empirically derived estimates [7–12] but lower than assumed by Peto *et al.* [1]. Since the proportion of household transmission may vary in different communities, we calculated $R_{eff}$ over a range of values of $SAR_{home}$ while keeping $R_0$ fixed at 2.5 (figure 1*b*). Similar results could be obtained by varying the mean household size. This shows that, even when one-third of transmission is within households ($SAR_{home} = 70\%$), mass testing and quarantine reduce $R_{eff}$ to 1.6 at best.

For simplicity, we ignored individual heterogeneity in non-household transmission. To test this assumption, we simulated the model with a negative binomial distribution for the number of non-household secondary cases, with values of the overdispersion parameter $k = 1$, 0.5 and 0.1 [13]. These values are in the range of estimates for COVID-19 transmission [14–16]. Individual heterogeneity modelled in this way affects the dynamics of small outbreaks and increases the probability of extinction for an outbreak from a single seed case at a given reproduction number [13]. However, our focus here is on the effect of a particular control measure on a large epidemic, in which case the effective reproduction number is the most important measure. Although heterogeneity increases

**Table 2.** Model parameter values, data sources and references.

| parameter | value | | | | | | | | Source |
|---|---|---|---|---|---|---|---|---|---|
| distribution of generation times (days) | $1.3 +$ Weibull (scale $= 5.67$, shape $= 2.83$) | | | | | | | | [17,18] |
| basic reproduction number | $R_0 = 2.5$ | | | | | | | | assumed |
| relative infectiousness of subclinical cases | $c_{sub} = 0.5$ | | | | | | | | [19,20] |
| relative onward transmission from isolated or quarantined cases | $c_{isol} = 0$ | | | | | | | | perfect isolation and quarantine assumed |
| proportion of subclinical infections | $p_{sub} = 0.33$ | | | | | | | | [20] |
| relative test sensitivity for subclinical infections | $p_{testsubclin} = 0.65$ | | | | | | | | [21] |
| household secondary attack rate | $SAR_{home} = 0.2$ (other values simulated in figure 1b) | | | | | | | | [12] |
| time from testing to return of test result | $t_{test} = 1$ day | | | | | | | | estimated |
| test sensitivity (probability of an infected individual testing positive) as a function of the number of days (0–38) since infection. Sensitivity is assumed to be zero more than 38 days after infection. | 0 | 1 | 2 | 3 | 4 | 5 | 6 | | [3] |
| | 0.00 | 0.00 | 0.01 | 0.04 | 0.33 | 0.62 | 0.75 | | |
| | 7 | 8 | 9 | 10 | 11 | 12 | 13 | | |
| | 0.79 | 0.80 | 0.79 | 0.76 | 0.72 | 0.68 | 0.64 | | |
| | 14 | 15 | 16 | 17 | 18 | 19 | 20 | | |
| | 0.60 | 0.56 | 0.52 | 0.48 | 0.44 | 0.40 | 0.37 | | |
| | 21 | 22 | 23 | 24 | 25 | 26 | 27 | | |
| | 0.34 | 0.32 | 0.30 | 0.28 | 0.26 | 0.24 | 0.22 | | |
| | 28 | 29 | 30 | 31 | 32 | 33 | 34 | | |
| | 0.20 | 0.18 | 0.16 | 0.14 | 0.12 | 0.10 | 0.08 | | |
| | 35 | 36 | 37 | 38 | | | | | |
| | 0.06 | 0.04 | 0.02 | 0.00 | | | | | |
| proportion of people living in a household of size 1 to 8 | 1 | 2 | 3 | 4 | 5 | 6 | 7 | 8 | [4] |
| | 17% | 36% | 19% | 17% | 7% | 3% | 0.7% | 0.6% | |

variability in individual reproduction numbers, we found that $R_{eff}$ was insensitive to introducing heterogeneity and varying the value of $k$ (table 1) [13]. Including age-specific contact patterns would allow the spread of the outbreak through different age groups to be modelled, but is unlikely to dramatically change the impact of the interventions examined on $R_{eff}$.

Rapid and widely available testing is undoubtedly a critical part of the response to COVID-19. However, it is dangerous to assert that particular interventions will be sufficient to control the spread of the virus without supporting evidence. A simple model shows that mass weekly testing and household quarantine, even if it were perfectly achievable, would not be sufficient to control the spread of COVID-19. This is due to a combination of significant pre-symptomatic transmission [22,23], low test sensitivity prior to symptom onset [3], delays in processing tests and returning results, and substantial non-household transmission. This implies that other measures would also need to be added. Given the logistical barriers to a programme of population-wide weekly testing, focusing

resources on more targeted controls, such as symptom-based testing combined with rapid tracing and quarantining of contacts, is preferable to an indiscriminate mass testing approach.

## 2. Methods

Table 2 shows data sources, parameter values and references. We assume that in the absence of any interventions $R_0 = 2.5$. This is split into non-household transmission $R_\text{out}$ and household transmission $R_\text{home} = \text{SAR}_\text{home} \bar{H}$ where $\text{SAR}_\text{home}$ is the household secondary attack rate and $\bar{H} = 1.74$ is the average number of household contacts [4]. Each household is randomly assigned a size according to the proportion of the population living in a household of size 1 to 8 in England and Wales [4] (2011 census, all usual residents aged 18 and over in households). In the absence of interventions, each clinical case has a probability $\text{SAR}_\text{home}$ of infecting each other member of the household. Each clinical case also infects an independent Poisson distributed number of people outside the household with mean $R_\text{out}$. To model individual heterogeneity in transmission rates, we ran simulations where the number of secondary cases outside the household was instead drawn from a negative binomial distribution with mean $R_\text{out}$ and overdispersion parameter $k$. We assume no individual can be re-infected, so each household transmission reduces the susceptible pool within that household. New cases infected outside the household are all assigned to new households, which are assumed to be fully susceptible initially. Times of secondary infections are distributed according to a Weibull distribution, time-shifted such that 35% of onward transmission occurs before symptom onset. We assume 33% of infections are subclinical and these are 50% as infectious as clinical cases and have 65% test sensitivity.

To simulate mass testing, we assume that everyone is tested weekly and has a positive test probability that depends on the time since infection [3]. This probability is very low in the first 3 days after infection and increases to a peak of 80% in 8 days after infection. We assume that more than 21 days after infection (the longest time with available data), the positive test probability continues to decrease by 2 percentage points per day (table 2). We assume that it takes one day for the test result to be returned and, following a positive test result, all subsequent transmission from the positive case and, in the household quarantine scenario, all their household members is prevented. If the test is a false negative, the individual is retested 7 days later with an independent probability of testing positive.

Data accessibility. This article has no additional data.

Competing interests. We declare we have no competing interests.

Funding. The authors were funded by Te Pūnaha Matatini, New Zealand's Centre of Research Excellence in Complex Systems, and the New Zealand Ministry of Business, Innovation and Employment.

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
