## [Peer Review File · Royal Society Open Science]

Review History

RSOS-201546.R0 (Original submission)

Review form: Reviewer 1

Is the manuscript scientifically sound in its present form?

Yes

Are the interpretations and conclusions justified by the results?

Yes

Is the language acceptable?

Yes

Do you have any ethical concerns with this paper?

No

Have you any concerns about statistical analyses in this paper?

No

Recommendation?

Accept as is

Comments to the Author(s)

A well written article with concise methods and results.

Review form: Reviewer 2 (Samuel Clifford)**Is the manuscript scientifically sound in its present form?**

Yes

Are the interpretations and conclusions justified by the results?

Yes

Is the language acceptable?

Yes

Do you have any ethical concerns with this paper?

No

Have you any concerns about statistical analyses in this paper?

Yes

Recommendation?

Accept with minor revision (please list in comments)

Comments to the Author(s)

Overall I am satisfied that the authors have provided a thorough analysis of the evidence in making their claims that weekly PCR testing (and perhaps antigen testing, too) might not be enough to control a local COVID-19 epidemic.

The authors discount the individual-level heterogeneity that can be modelled by using an overdispersed distribution. While they are correct that the mean reproduction number won't change, a simulation study sampling $R_i \sim f\left(R_0, k\right)$ for each individual would result in a variable number of ongoing transmission events per individual, and stochastically sampling which individuals quarantine effectively would likely lead to a different mean R_{eff} and certainly an informative estimate of uncertainty around it. Endo et al. (2020) estimate that for $R_0 = 2.5$ we have $k=0.1$ which indicates that approximately 80% of secondary transmission is caused by 10% of infectious individuals. Wang et al. (2020) estimate $k=0.23$ and Riou and Althaus provide an early estimate of $k=0.54$; these are not as overdispersed but certainly may drive deviation from assumption with fixed R . I would like to see the authors do a sensitivity analysis on the assumption of Poisson versus Negative Binomial with overdispersion drawn from the literature.

Endo et al. (2020) <https://wellcomeopenresearch.org/articles/5-67/v3>

Wang et al. (2020) <https://www.nature.com/articles/s41467-020-18836-4>

Riou and Althaus (2020) <https://www.eurosurveillance.org/content/10.2807/1560-7917.ES.2020.25.4.2000058>

Decision letter (RSOS-201546.R0)

Dear Mr Steyn

On behalf of the Editors, we are pleased to inform you that your Manuscript RSOS-201546 "Comment: Weekly COVID-19 testing with household quarantine and contact tracing is feasible and would probably end the epidemic" has been accepted for publication in Royal Society Open Science subject to minor revision in accordance with the referees' reports. Please find the referees' comments along with any feedback from the Editors below my signature.

Please submit your revised manuscript and required files (see below) no later than 7 days from today's (ie 28-Jan-2021) date. Note: the ScholarOne system will 'lock' if submission of the revision is attempted 7 or more days after the deadline. If you do not think you will be able to meet this deadline please contact the editorial office immediately.

on behalf of the Associate Editor, and Professor Mark Chaplain (Subject Editor)
openscience@royalsociety.org

Reviewer comments to Author:
Reviewer: 1
Comments to the Author(s)

A well written article with concise methods and results.

Reviewer: 2
Comments to the Author(s)

Overall I am satisfied that the authors have provided a thorough analysis of the evidence in making their claims that weekly PCR testing (and perhaps antigen testing, too) might not be enough to control a local COVID-19 epidemic.

The authors discount the individual-level heterogeneity that can be modelled by using an overdispersed distribution. While they are correct that the mean reproduction number won't change, a simulation study sampling $R_i \sim f\left(R_0, k\right)$ for each individual would result in a variable number of ongoing transmission events per individual, and stochastically sampling which individuals quarantine effectively would likely lead to a different mean R_{eff} and certainly an informative estimate of uncertainty around it. Endo et al. (2020) estimate that for $R_0 = 2.5$ we have $k=0.1$ which indicates that approximately 80% of secondary transmission is caused by 10% of infectious individuals. Wang et al. (2020) estimate $k=0.23$ and Riou and Althaus provide an early estimate of $k=0.54$; these are not as overdispersed but certainly may drive deviation from assumption with fixed R_0 . I would like to see the authors do a sensitivity analysis on the assumption of Poisson versus Negative Binomial with overdispersion drawn from the literature.

Endo et al. (2020) <https://wellcomeopenresearch.org/articles/5-67/v3>

Wang et al. (2020) <https://www.nature.com/articles/s41467-020-18836-4>

Riou and Althaus (2020) <https://www.eurosurveillance.org/content/10.2807/1560-7917.ES.2020.25.4.2000058>

===PREPARING YOUR MANUSCRIPT===

If you have been asked to revise the written English in your submission as a condition of publication, you must do so, and you are expected to provide evidence that you have received language editing support. The journal would prefer that you use a professional language editing service and provide a certificate of editing, but a signed letter from a colleague who is a native speaker of English is acceptable. Note the journal has arranged a number of discounts for authors

using professional language editing services
(<https://royalsociety.org/journals/authors/benefits/language-editing/>).

===PREPARING YOUR REVISION IN SCHOLARONE===

-- If you have uploaded ESM files, please ensure you follow the guidance at <https://royalsociety.org/journals/authors/author-guidelines/#supplementary-material> to include a suitable title and informative caption. An example of appropriate titling and captioning may be found at https://figshare.com/articles/Table_S2_from_Is_there_a_trade-

off_between_peak_performance_and_performance_breadth_across_temperatures_for_aerobic_sc
ope_in_teleost_fishes_/3843624.

Author's Response to Decision Letter for (RSOS-201546.R0)

See Appendix A.

RSOS-201546.R1 (Revision)

Review form: Reviewer 2 (Samuel Clifford)

Is the manuscript scientifically sound in its present form?

Yes

Are the interpretations and conclusions justified by the results?

Yes

Is the language acceptable?

Yes

Do you have any ethical concerns with this paper?

No

Have you any concerns about statistical analyses in this paper?

No

Recommendation?

Accept as is

Comments to the Author(s)

I'd like to thank the editor for the opportunity to re-review this article. In my previous review I had mentioned that the authors should perform a sensitivity analysis on the possible overdispersion in R among the individuals in the model. I am grateful to the authors for performing this and citing the relevant literature, and was interested to see that the result for R_{eff} was mostly insensitive to k ; I have no further concerns about the article and recommend its publication.

Decision letter (RSOS-201546.R1)

Dear Mr Steyn,

It is a pleasure to accept your manuscript entitled "Comment: Weekly COVID-19 testing with household quarantine and contact tracing is feasible and would probably end the epidemic" in its current form for publication in Royal Society Open Science. The comments of the reviewer(s) who reviewed your manuscript are included at the foot of this letter.

COVID-19 rapid publication process:

We are taking steps to expedite the publication of research relevant to the pandemic. If you wish, you can opt to have your paper published as soon as it is ready, rather than waiting for it to be published the scheduled Wednesday.

This means your paper will not be included in the weekly media round-up which the Society sends to journalists ahead of publication. However, it will still appear in the COVID-19 Publishing Collection which journalists will be directed to each week (<https://royalsocietypublishing.org/topic/special-collections/novel-coronavirus-outbreak>).

If you wish to have your paper considered for immediate publication, or to discuss further, please notify openscience_proofs@royalsociety.org and press@royalsociety.org when you respond to this email.

Kind regards,
Anita Kristiansen

Editorial Coordinator

on behalf of Mark Chaplain (Subject Editor)
openscience@royalsociety.org

Reviewer comments to Author:
Reviewer: 2

Comments to the Author(s)

I'd like to thank the editor for the opportunity to re-review this article. In my previous review I had mentioned that the authors should perform a sensitivity analysis on the possible overdispersion in R among the individuals in the model. I am grateful to the authors for performing this and citing the relevant literature, and was interested to see that the result for R_{eff} was mostly insensitive to k ; I have no further concerns about the article and recommend its publication.

Appendix A

Author response (bold text) to Reviewer comments

Reviewer: 1

Comments to the Author(s)

A well written article with concise methods and results.

Thank you.

Reviewer: 2

Comments to the Author(s)

Overall I am satisfied that the authors have provided a thorough analysis of the evidence in making their claims that weekly PCR testing (and perhaps antigen testing, too) might not be enough to control a local COVID-19 epidemic.

The authors discount the individual-level heterogeneity that can be modelled by using an overdispersed distribution. While they are correct that the mean reproduction number won't change, a simulation study sampling $R_i \sim f\left(R_0, k\right)$ for each individual would result in a variable number of ongoing transmission events per individual, and stochastically sampling which individuals quarantine effectively would likely lead to a different mean R_{eff} and certainly an informative estimate of uncertainty around it. Endo et al. (2020) estimate that for $R_0 = 2.5$ we have $k=0.1$ which indicates that approximately 80% of secondary transmission is caused by 10% of infectious individuals. Wang et al. (2020) estimate $k=0.23$ and Riou and Althaus provide an early estimate of $k=0.54$; these are not as overdispersed but certainly may drive deviation from assumption with fixed R . I would like to see the authors do a sensitivity analysis on the assumption of Poisson versus Negative Binomial with overdispersion drawn from the literature.

This is a good point – we did neglect individual heterogeneity. We have now a sensitivity analysis (see Table 2) with different levels of individual heterogeneity, as measured by the dispersion parameter $k = 1, 0.5$ or 0.1 , motivated by the suggested references. Although this adds variation in individual reproduction numbers, we find that the effective reproduction number is insensitive to the introduction of heterogeneity and we have added a brief discussion of this (lines 51-60 of page 1 and continuing on lines 3-12 of page 2).

Endo et al. (2020) <https://wellcomeopenresearch.org/articles/5-67/v3>

Wang et al. (2020) <https://www.nature.com/articles/s41467-020-18836-4>

Riou and Althaus (2020) <https://www.eurosurveillance.org/content/10.2807/1560-7917.ES.2020.25.4.2000058>